# Single-molecule electrical contacts on silicon electrodes under ambient conditions

Albert C. Aragonès[1,2,3], Nadim Darwish[4], Simone Ciampi[4], Fausto Sanz[1,2,3], J. Justin Gooding[5] & Ismael Díez-Pérez[1,2,3]

The ultimate goal in molecular electronics is to use individual molecules as the active electronic component of a real-world sturdy device. For this concept to become reality, it will require the field of single-molecule electronics to shift towards the semiconducting platform of the current microelectronics industry. Here, we report silicon-based single-molecule contacts that are mechanically and electrically stable under ambient conditions. The single-molecule contacts are prepared on silicon electrodes using the scanning tunnelling microscopy break-junction approach using a top metallic probe. The molecular wires show remarkable current–voltage reproducibility, as compared to an open silicon/nano-gap/metal junction, with current rectification ratios exceeding 4,000 when a low-doped silicon is used. The extension of the single-molecule junction approach to a silicon substrate contributes to the next level of miniaturization of electronic components and it is anticipated it will pave the way to a new class of robust single-molecule circuits.

[1] Department of Materials Science and Physical Chemistry & Institute of Theoretical and Computational Chemistry (IQTC), University of Barcelona, Martí i Franquès 1, 08028 Barcelona, Spain. [2] Institute for Bioengineering of Catalonia (IBEC), Baldiri Reixac 15-21, 08028 Barcelona, Spain. [3] Centro Investigación Biomédica en Red (CIBER-BBN), Campus Río Ebro-Edificio I + D, Poeta Mariano Esquillor s/n, 50018 Zaragoza, Spain. [4] Department of Chemistry, Faculty of Science & Engineering, Curtin University, Nanochemistry Research Institute, Perth, Western Australia 6102, Australia. [5] School of Chemistry and Australian Centre for NanoMedicine, The University of New South Wales, Sydney, New South Wales 2052, Australia. Correspondence and requests for materials should be addressed to N.D. (email: nadim.darwish@curtin.edu.au) or to I.D.-P. (email: isma_diez@ub.edu).

The metal–semiconductor junction is a pillar of the microelectronic industry. There are several technical barriers to a further miniaturization of the current metal–semiconductor circuitry, and it is hoped that these obstacles may be removed by integrating individual organic molecules to form equivalent metal/molecule/semiconductor junctions. In addition to facilitating miniaturization, this type of molecular junctions would enable utilizing the incredibly diverse range of chemistries that currently exist to expand the electrical properties of inorganic solid state devices[1]. Hence, by combining the electrical properties of semiconductors with the chemical diversity of organic molecules, a variety of device functionalities can be realized.

Traditionally, the idea of using single molecules in electronics has been focusing on molecules bonded between two metal electrodes[2–6]. Recently, there have been considerable advances in the chemical aspects of molecular electronics on semiconducting surfaces[7–11]. In industry, the dominance of semiconductors, mainly silicon, suggests the development of a new generation of electronics, where thin films or even single molecules are incorporated in the conventional technology which is based on silicon. This development will be the most likely commercial development of even smaller electronic devices. Hence, an effort to bridge the gap between current single-molecule technology and silicon circuitry appears to be an unavoidable step towards the near future of the electronics field[12].

Most efforts to date to prepare metal/molecule/silicon electrical contacts have involved macroscopic devices (ca. $10^{-3}\,\text{cm}^2$) comprising very large ensembles of molecules[13–16]. The top contact with a second electrode is usually made via metal deposition on the top of the organic layer either by evaporation or via Hg or alloy-metal contacts[17,18]. This macroscopic approach has provided most of the knowledge we have about charge transport across the molecule-semiconductor junction, however, they do suffer from critical drawbacks as fundamental platforms for charge transport studies. For instance, it is known that deposition of gold films onto the distal end of molecular monolayers assembled on silicon surfaces causes penetration of the metal into the monolayer (pinholes) resulting in deteriorated performance and reproducibility[19,20]. There are also issues regarding the quality of the contact between the top electrode and the organic layer that has enormous effects on the current responses. For instance, current–voltage properties of a large-scale tunnelling junctions can range from those of a resistor to those of a diode depending on the size and geometry of the top-metal contact[21,22]. Hence, fundamental approaches to study metal/molecule/silicon electrical contacts go through controlling the physical and chemical properties of the top contact between the metal electrode and the monolayer-covered silicon electrode. One of the solutions would be a contact between the top-metal electrode and an individual molecule connected to the bottom silicon electrode, which allows for a fixed distance between the junction electrodes and would limit to one (the molecular backbone) the channels across which electrons transport.

Over the last decade, there have been great technical advances in electrical measurements at the single-molecule level[2,3,6,23] and, for instance, we and others have been able to assemble single-molecule diodes by bridging chemically asymmetrical molecules between two metal leads[24–26]. Such single-molecule circuits constitute the prove of concept of single-molecule circuitry, although they have been limited to metal platforms leading to a restricted scope for practical applications due to their mechanical instability, their limited rectification ratios and their sensitivity to the structure of the molecule-metal contact[24].

In this article, we take advantage of recent advances in the surface chemistry of hydrogenated silicon surfaces[7,8,27,28] and in the scanning tunnelling microscope break junction (STM-BJ) approach to report on a silicon-based single-molecule contacts that operates under ambient conditions with high rectification ratios and high mechanical stability. We show that the molecular contact enables good control of the metal/silicon gap distance under ambient conditions thus enhancing the reproducibility of its electrical properties avoiding the necessity of ultra-high vacuum and/or cryogenic conditions[29,30]. This approach also offers a fundamental platform to study molecular charge transport addressing the variability issue of large area metal–organic–semiconductor measurements. In this work, a symmetrical saturated alkyl chain molecule that comprise two terminal alkyne groups is chemisorbed on a Si(−111)-H surface. This chemical approach protects the silicon against oxidation and provides a distal alkyne unit, which has high affinity to a metallic STM tip that serves as the top contact of the device. A gold STM tip is then used to wire a single-nonadiyne molecule between the gold and n-type silicon electrodes via the single-molecule junction technique. Current–voltage measurements of single-nonadiyne molecules are then performed and the dependence of the rectification ratios on the silicon doping level is evaluated. Using this silicon-molecule-metal approach, we have built a molecular-scale diode of controllable rectification ratio that can reach exceedingly high values up to 4,000.

## Results

**Functionalizing the Si(−111)-H surface.** With the purpose of forming single-molecule junctions between gold and silicon electrodes under ambient conditions, it was first important to choose molecules that can both prevent the silicon from oxidation, thus preserving its semiconducting properties, as well as being able to form a stable contact at a fixed distance between silicon and gold electrodes. This reduces the ambiguity as to where the electron transport process is occurring in large ensembles of molecules and fixes the distance between the two junction electrodes. Monolayers of 1,8-nonadiyne (Fig. 1) meet these requirements and have been used to passivate silicon surfaces with highly ordered monolayers via hydrosilylation reactions[7,8,10,27,31].

The 1-alkyne at the distal end of the monolayer was previously shown to form stable contacts with macroscopic gold electrodes[32,33], hence it is perfectly suited to form an electrical contact with a gold STM tip. Figure 1b shows a $5 \times 5\,\mu\text{m}^2$ atomic force microscopy (AFM) image of a nonadiyne-modified n-type

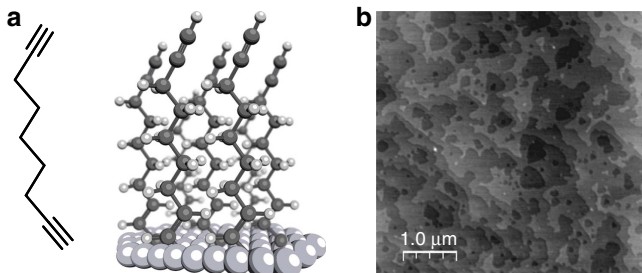

**Figure 1 | Self-assembled monolayer of nonadiyne on Si(−111) electrodes.** (**a**) Chemical structure of 1,8-nonadiyne alongside a schematic of the monolayer formed from 1,8-nonadiyne on H-terminated silicon via ultraviolet-assisted hydrosilylation reaction. The reaction was allowed to take place overnight under an inert atmosphere (**b**) (5 μm × 5 μm) AFM image, in tapping mode, at ambient conditions of an H-terminated n-type Si(−111) surface modified with 1,8-nonadiyne showing that the atomically flat Si(−111) terraces are preserved after the formation of the monolayer.

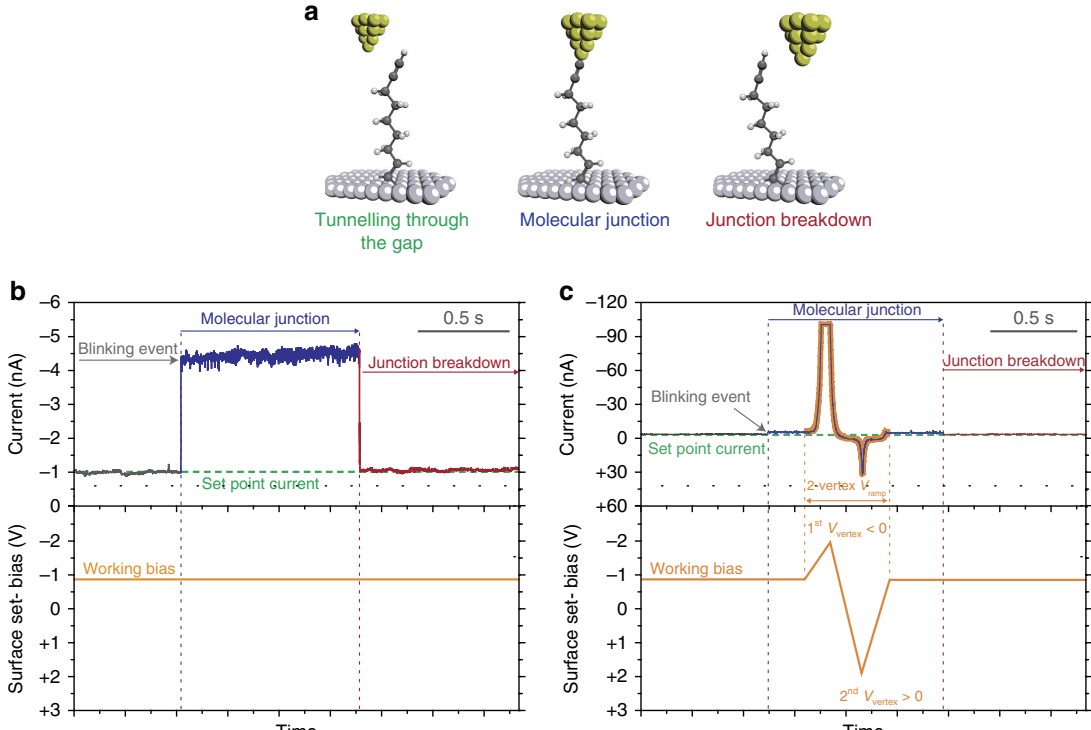

**Figure 2 | STM break-junction experiments of gold–nonadiyne–silicon junctions.** (**a**) Schematic of the STM junction experiment describing the formation and breakdown of the Au–nonadiyne–Si junction. (**b**) Example of a captured 'blink' when a 1,8-nonadiyne molecule bridges the two electrodes at $-0.8$ V (see blinks at different biases in Supplementary Fig. 1 and Supplementary Note 1). Current jumps (blinks) above the set-point tunnelling current appear when the nonadiyne molecule bridges the gap between the gold and the silicon electrodes. (**c**) Captured current response during a voltage bias ramp of an Au–nonadiyne–Si$_{HD}$ junction (see the current response for the low-doped silicon system, Au–nonadiyne–Si$_{LD}$, in Supplementary Fig. 2). When a blink is detected, the working bias was ramped between $+2$ and $-2$ V at 2 Hz of frequency (16 V s$^{-1}$). The current is continuously monitored after the bias ramp to ensure that the junction does not breakdown during the voltage ramp. Surface set-bias plots of **b,c** represent the evolution of the applied working bias voltage between the Si substrate and the STM Au-tip electrodes.

Si($-111$). The topography shows flat terraces separated by atomic steps. The peak-to-peak roughness measured within one Si($-111$) terrace is ca. 0.15 nm which is consistent with an atomically smooth alkyl monolayer on Si($-111$) (ref. 10). More importantly, the high-quality topography of the alkyne monolayers confirms that our procedure yields homogeneous monolayers at the nanoscale free of any contaminants or oxidation. This agrees with XPS measurements (Supplementary Fig. 11) that showed no silicon oxides or sub-oxides (102–104 eV)[34], which is indicative of a well-formed monolayer[35–38]. Furthermore, the presence of XPS C 1s signals corresponding to –C≡C– groups (286.7 eV) and to a silylated olefin (Si–C=C, 283.9 eV) is another indication of the effective formation of a monolayer of nonadiyne on the silicon surface[39–41].

**Connecting individual molecules.** After confirming the quality of the nonadiyne monolayer on the silicon electrodes, we used the blinking approach of the STM junction technique to wire single molecules with a gold STM tip[23,24,42,43]. In a typical blinking experiment, an initial set-point tunnelling current is applied under a fixed bias voltage, therefore the STM tip and the alkyne-terminated silicon electrode are fixed at a specific distance. After reaching the current set-point, the feedback loop of the current is turned off and the current is recorded as a function of time (Fig. 2a,b). Current jumps above the tunnelling current (blinking events) appear when a nonadiyne molecule occupies the gap between the gold and the silicon electrodes (Fig. 2b). The

formation of mechanically stable molecular junctions was confirmed by collecting mechanical pulling curves during the 'blinks' (Fig. 3)[23]. The pulling is based on the application of a vertical tip retraction stage using the piezoelectric translator attached to the STM tip, increasing the gap distance between both electrodes. Pulling curves collected during the lifetime of the 'blinks' showed plateaus with a pulling length of ca. 0.3 nm (Fig. 3a), which resembles the stretching of the junction[24,42]. When the pulling stimulus was applied in the absence of blinks (that is, over the tunnelling background, Fig. 3b), an event-free exponential decay was observed instead, confirming that the blinks are a consequence of a mechanically stable molecular junctions[23,43]. In a series of control experiments, no evidence of blinks was observed when the 1,8-nonadiyne monolayer was changed to another prepared from 1-nonyne on Si($-111$)–H. Monolayers of 1-nonyne are methyl-terminated and lack the distal alkyne moiety that is necessary to form an effective electrical contact to the STM gold tip (Fig. 5b and Supplementary Fig. 9).

**Current–voltage properties of the single-molecule junctions.** Current–voltage (I(V)) characteristics (Fig. 4a) were collected during the timescale when the molecule is attached between the electrodes (Fig. 2c and Supplementary Fig. 2) in the absence of light. The I(V) of the single-molecule junctions measured on the highly n-doped silicon (0.003–0.007 Ω cm) shows very low current rectification (red curve). The highly doped

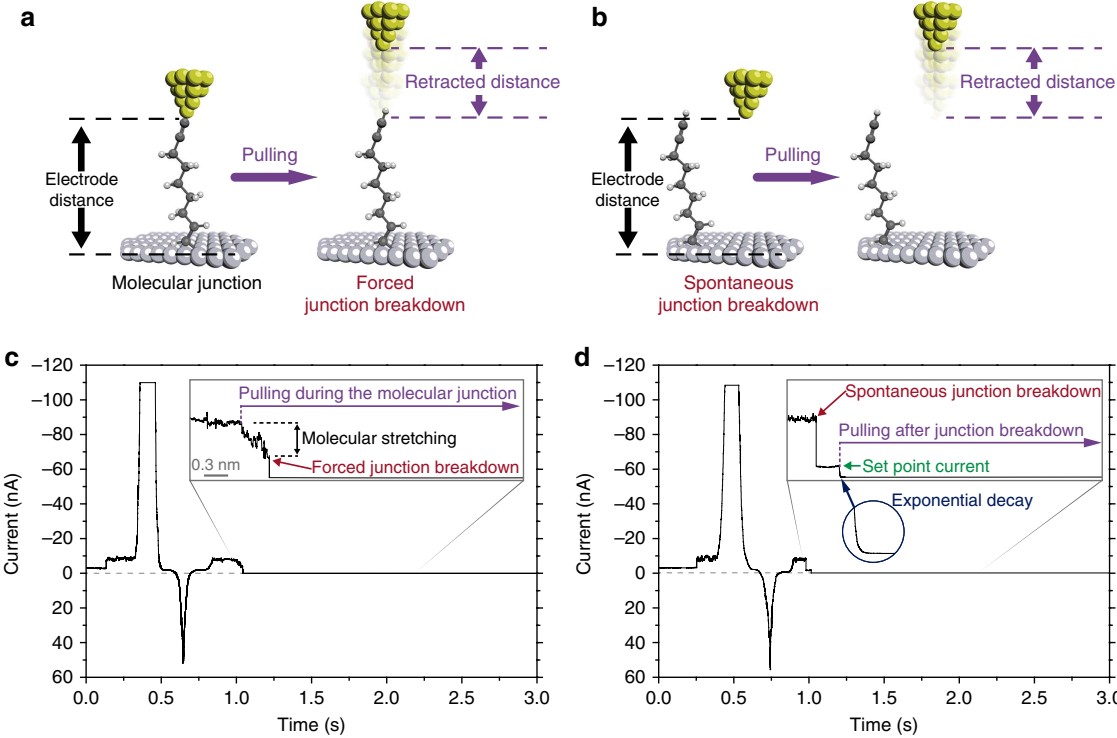

**Figure 3 | Pulling single-molecule junctions.** (**a**) Schematic representation of the pulling stage over a blinking event, before the spontaneous junction breakdown occurs. (**b**) Schematic representation of the pulling stage after the spontaneous junction breakdown occurred. (**c**) Pulling curves exerted before the junction breakdown (**a**) showing mechanical resistance confirming the molecular signature of the junction and its stability during the tip retraction. (**d**) Pulling curves exerted after the spontaneous junction breakdown occurred (**b**) showing immediate exponential decay without mechanical resistance suggesting the absence of a molecular junction at this stage. The initial set point is 1 nA.

silicon, here after referred to as (Si$_{HD}$), has a doping density of $N_D \sim 10^{19}$ cm$^{-3}$ (ref. 44). This silicon is a quasi-degenerated semiconductor with charge carrier densities that are closer to a metal doping level ($N_D \sim 10^{22}$ cm$^{-3}$) than they are to an intrinsic semiconductor ($N_D \sim 10^{10}$ cm$^{-3}$). Figure 4c shows the energy band diagram representations of the different charge transport regimes along the I(V) characteristic in the Au–nonadiyne–Si junctions. Under accumulation conditions (Fig. 4c, left panel), charge carriers (electrons for an n-type semiconductor) will concentrate on the n-Si$_{HD}$ electrode surface, which will easily tunnel through the organic molecular backbone at the applied negative bias voltage (negative I(V) branch). Oppositely, under depletion conditions (Fig. 4c, central panel), a thin space charge region (SCR) will develop on the n-Si$_{HD}$ electrode surface due to the high charge carriers' concentration. Using the Schottky relation to evaluate the thickness of the SCR in the Si$_{HD}$ (ref. 45), we get values below 10 nm by taking a surface energy barrier of 1 V corresponding approximately to the threshold voltage observed in the Au–nonadiyne–Si$_{HD}$ I(V) (Fig. 4a). Solving the Poisson's equation that describes the electric field across the semiconductor[45], we get a parabolic potential profile across the SCR (Fig. 4c), which implies a much narrower net barrier, $\sim 1$ nm, at the injecting potential of the Au STM tip electrode close to the band edge (Fig. 4c, central panel). Such barrier thickness is below the molecular length and the charge transfer process at the Au–nonadiyne–Si$_{HD}$ junction will be dominated by the electron tunnelling through the barrier imposed by the molecular frontier orbital[46], with small contribution from the semiconductor SCR. This results in the observed current flow at positive bias voltages (positive I(V) branch in the red curve of Fig. 4a). Contrarily, the I(V) curves measured with the low-doped silicon (Si$_{LD}$) silicon (1–10 Ω cm,

$N_D \sim 10^{15}$ cm$^{-3}$)[44] are asymmetrical around the origin with only a very low leakage current of 0.02 nA when the silicon surface is positively biased, in contrast to the large currents detectable when the surface is negatively biased (green curve of Fig. 4a). The strong current rectification is typical of n-type semiconductor, and indicates an increased barrier for the injected current from the STM tip contact towards the silicon surface (Fig. 4c, right panel). Following a similar reasoning, while the n-Si$_{LD}$ still provides with available charge carriers (electrons) at the semiconductor surface under accumulation conditions (negative I(V) branch), a thick SCL is expected to develop under depletion conditions due to the much lower charge carriers concentration on the semiconductor surface. An estimation of the SCR thickness in the Si$_{LD}$ yields values of $\sim 3$ μm, which leads to thick tunnelling barriers in the order of hundreds of nm at the Au tip-injecting potential. The development of such thick SCR in the low n-doped semiconductor surface under depletion conditions (positive I(V) branch in the green curve of Fig. 4a) is responsible for the electron tunnelling blockage, causing the observed large rectification behaviour of the single-molecule junction. Most notably, the rectification ratio for the junction is close to 4,000 (the ratio of forward to reverse current at 1.5 V), higher than any single-molecule diode previously reported. In all I(V) recordings, no hysteresis was observed in both forward and backward directions of the voltage bias scan (Supplementary Fig. 3), ruling out any significant charging effect at the employed voltage scan rate ($\sim 16$ V s$^{-1}$).

**Characterizing junctions in the absence of molecules**. I(V) curves were also collected in the absence of the nonadiyne molecules (Fig. 4b and Supplementary Fig. 6a,b). Hence

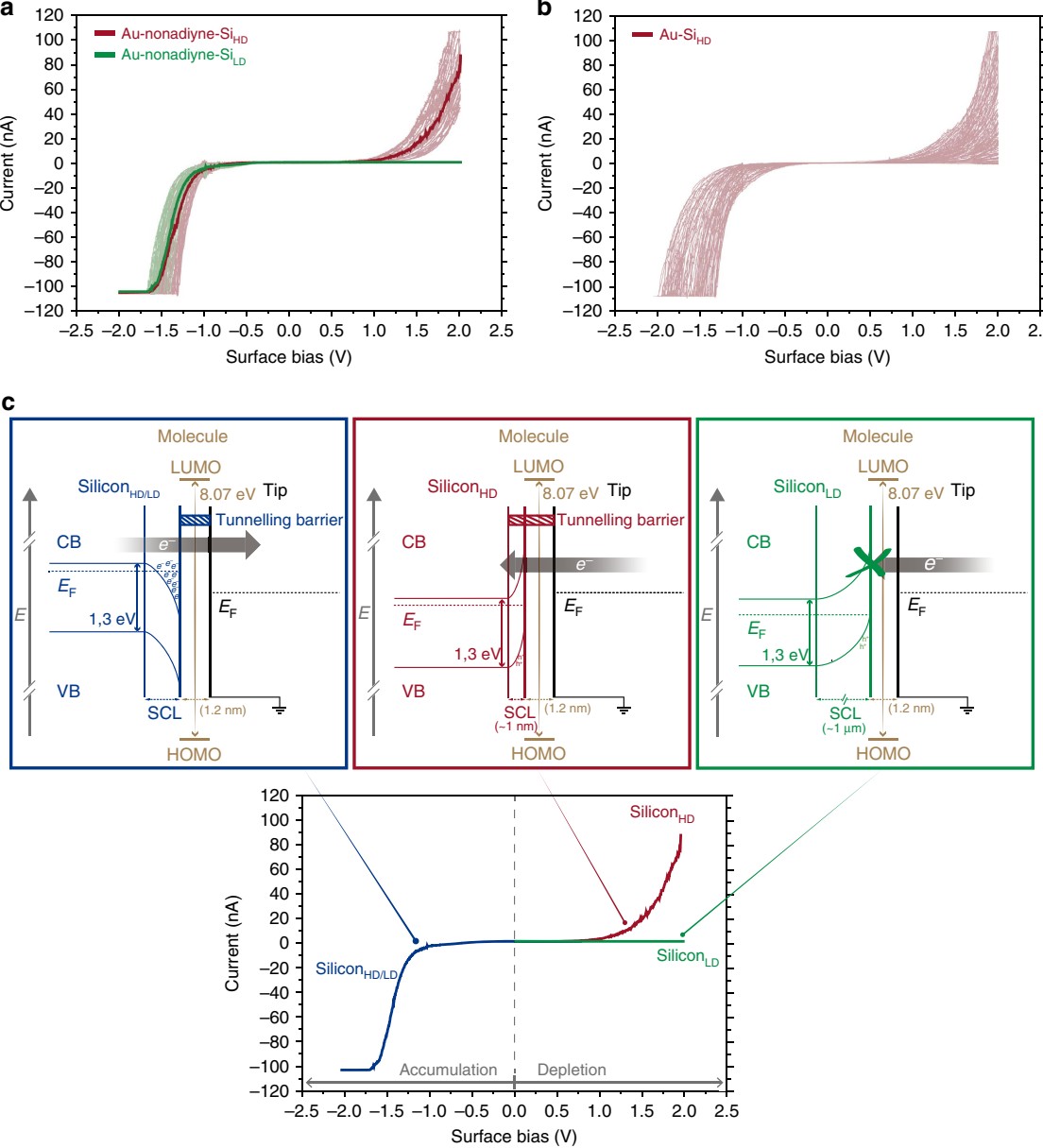

**Figure 4 | Current–voltage characteristics of gold–nonadiyne–silicon junctions.** (**a**) Current–voltage properties of single-nonadiyne junctions of the form Au–nonadiyne–Si$_{HD}$ (red curves) and Au–nonadiyne–Si$_{LD}$ (green curves). The solid lines are the average of all the plotted curves (35 curves for both Si$_{HD}$ and Si$_{LD}$). In the absence of a molecule (see **b**), the coefficient of variation ($C_V$) increases eightfold for the low-doped silicon (Si$_{LD}$) case and sixfold for the high-doped silicon, Si$_{HD}$ (Supplementary Table 1). (**b**) Current–voltage characteristics of an Au-gap–Si$_{HD}$ junction (35 curves). The bias sweep starts and ends at $-0.8$ V for all I(V) curves collected (Supplementary Fig. 4) (**c**) Energy band diagrams representations of the Au–nonadiyne–Si junction under charge carriers' accumulation conditions (applied negative sample voltages) in the Si electrode (left panel) and under depletion conditions (applied positive sample voltage) for the Si$_{HD}$ (central panel) and Si$_{LD}$ (right panel). Changing the starting bias does not affect the shape of the IV curves (Supplementary Fig. 5). The initial set point used is 1 nA. The computed bandgap energy of 1,8-nonadiyne is 8.07 eV (Supplementary Fig. 12).

the voltage was ramped at either a fixed position above a H-terminated silicon or at different X, Y or Z positions. It was found that a consistent I(V) response cannot be reproduced due to either variation in the tunnelling distance (Z direction) that is rapidly changing at room temperature, or a possible variation in the chemical functional groups on the silicon substrate (partial formation of SiOx due to the absence of molecular protection of the Si–H surface) in the XY direction. Unlike the case of a single-molecule junction with nonadiyne, the 'open-gap' tunnelling junctions showed a random and unpredictable rectification ratios regardless of the doping level of the

underneath silicon substrate (Fig. 4b and Supplementary Fig. 6a,b). In the absence of a molecule, the coefficient of variation ($C_V$) increases sixfolds in the absence of a molecule (for Si$_{HD}$ case) and eightfolds for the (Si$_{LD}$ case; Supplementary Table 1). The poor control on the electrode/electrode gap separation in the absence of molecule sometimes yields to physical contact of the silicon substrate by the Au STM tip, which dramatically changes the I(V) characteristics to that of an Ohmic contact regardless of the doping level used in the silicon (Supplementary Fig. 6c). This Ohmic behaviour is attributed to gold leaking from gold tip to the silicon surface thus forming gold–gold–silicon junctions.

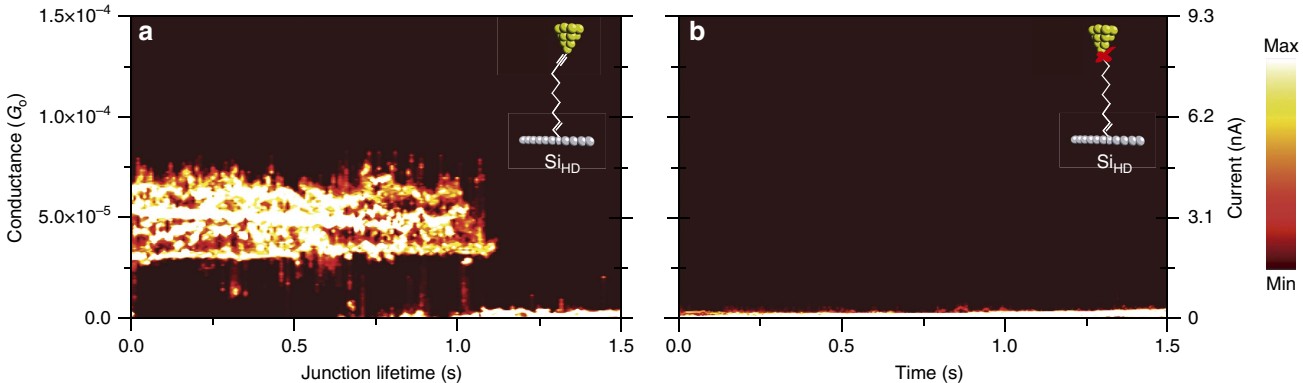

**Figure 5 | Blinking experiments of nonadiyne versus nonyne monolayers.** (**a**) 2-dimension blinking maps were built by the accumulation of hundreds of blinks at a common time $x$ axis origin. The two-dimension blinking maps were normalized versus the total amount of counts and a vertical offset to zero-current was applied to the baseline such that the single-molecule junction component are displayed clearly (tunnelling background subtracted). The average lifetime of the blinks is 0.75 s. (**b**) Two-dimension blinking maps with 1-nonyne monolayer showing the absence of blinks due to the lack of distal alkyne moieties (see Supplementary Fig. 9 for the low-doped silicon case). The initial set-point current and the applied surface-voltage bias are 1 nA and − 0.8 V, respectively.

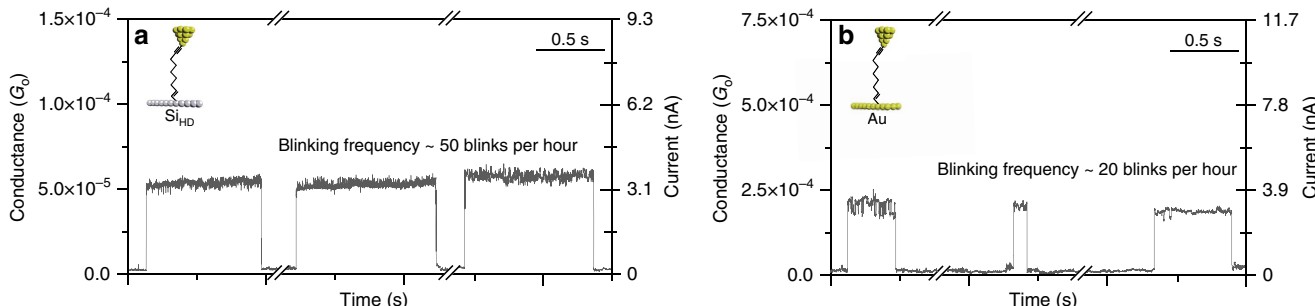

**Figure 6 | Blinking experiments of nonadiyne on different substrates.** Representative current traces showing blinks that correspond to the formation of single-molecule junctions of 1,8-nonadiyne molecules on Si (**a**) and gold (**b**) substrates. The $x$ axis does not represent a continuous acquisition time and time-breaks between each blink are added to account for this. A vertical offset has been applied to the baseline near to zero-current to display the single-molecule junction component only (tunnelling background subtracted). The applied surface-voltage biases are − 0.8 and − 0.2 V for Si and Au substrates, respectively, causing a current level for both type of junctions to be ∼3 nA. The initial set-point current was 1 nA in both cases.

This is supported by a similar Ohmic behaviour recorded for I(V) curves collected when the STM tip is in contact with a $Si_{LD}$ surface covered by a thin film of gold (∼25 nm) (Supplementary Fig. 6d). Such interfaces have been reported to behave as Ohmic contacts and is attributed to interfacial dipoles from high concentrations of interfacial charge or due to recombination centres created by interfacial Au in Si (refs 47–49). Hence, we conclude that a nonadiyne molecule in the junction is essential to prevent distance and/or chemical disparity between the junction electrodes, which leads to variations in the interfacial energetics that governs the current response of the nanoscale molecular device.

**Assessing the mechanical stability.** In order to assess the mechanical stability of the Au–nonadiyne–Si single-molecule junctions against previously reported metal/molecule/metal junctions, we have collected blinking captures at silicon surface-bias of − 0.8 V bias for both the $Si_{HD}$ (Fig. 5a) and $Si_{LD}$ systems (Supplementary Fig. 7). The dispersion in conductance values (between $3.45 \times 10^{-5}$ and $7.00 \times 10^{-5}$ $G_o$ where $G_o = 77.5 \, \mu S$) observed in Fig. 5a and (Supplementary Fig. 7) can be ascribed to intrinsic junction-to-junction variation in the molecule/gold electrode contact geometry during stretching[50,51]

or, in some instances, to the formation of multi-molecular bridges. Other source of conductance variation can be attributed to: (1) the uncertainty in the stereochemistry of the silylated olefin (reaction of Si–H with 1-alkynes). The produced silylated olefin could be either *trans* or *cis*[52] which could have different conductivities, and (2) the different bias applied (− 0.8 V for silicon versus − 0.2 V for gold to maintain the current for both cases in the ca. 3 nA range) which can induce more fluctuation in the silicon case, since the imposed electric field is much higher and can lead to the stabilization of higher energy configurations of the molecular junction. The average lifetime was found to be 0.75 s, which is ca. 30% higher than that with the same molecule but linked between two gold electrodes (Supplementary Fig. 10 and Supplementary Note 2). This enhanced junction lifetime is attributed to the strong covalent Si–C bond upon the ultraviolet-hydrosilylation reaction of nonadiyne with the H-terminated Si and it is independent on the doping level of the Si electrode. Another unique property of the single-molecule junctions formed on silicon is the enhanced frequency of the junction formation. The frequency of the junctions increases 2.5 folds (20 junction per hour to 50 junctions per hour) when a silicon substrate was used as the base electrode instead of a standard Au electrode (Fig. 6). This enhanced junction formation is attributed to the absence of mobility

of the nonadiyne monolayer on silicon as opposed to the high mobility of the thiol monolayer of gold electrodes. This is supported by occasionally observing more than one blink appearing in a 3 s frame (35% of the blinks in the silicon case appeared accompanied by other blinks versus 5% in the gold case). An example of a high-frequency blinking capture in a 3 s frame is shown in Supplementary Fig. 8. We speculate that the non-homogenous and sudden increase in the frequency of the junction in the silicon case is attributed to the rigidity of the carbon-silicon bond which enables the same single molecule to form a junction multiple times in short periods of time (breaking from the weak gold electrode and then reforming again) which would lead to a sudden increase in the blinking frequency.

## Discussion

We have demonstrated very high and reproducible current rectification ratios in a single-molecule electrical contact under ambient conditions. This single-molecule diode is achieved using silicon and gold materials as the junction electrodes and an alkyl chain molecule terminated with alkyne moieties. The single-nonadiyne molecule enables mechanically stable, chemically pure and distance-defined molecular junctions on silicon electrodes. These properties enable reproducible electrical responses and mechanical properties superior to that of their gold electrodes counterparts. The increased junction lifetime is important, as it is a prerequisite for the application of potential ramp while the molecule is still attached between the two electrodes. It is also an essential feature in the application of other *in-situ* stimuli on single molecules, such as light or voltage gate. The enhanced mechanical stability of single-molecule junctions on silicon electrodes is hoped to open the door for device applications where mechanical stability becomes the most critical issue in single-molecule electronics. Using the silicon-molecule-metal approach, we have built a molecular-scale diode of controllable rectification ratio that can reach an excess of 4,000 in the case when low-doped silicon is used. The integration of single-molecule contacts with semiconductors adds another dimension to tune the electron transport across single molecules, which can potentially contribute to a class of single-molecule electronics compatible with the conventional semiconducting circuitry.

## Methods

**Surface modification.** After cutting Si( − 111) wafer to a size appropriate for the STM cell, the wafer was sonicated in re-distilled DCM, acetone, ethanol and MilliQ water (18 MΩ cm$^{-1}$), 5 min in each solvent. The wafer was then oxidized in Piranha solution (1 $H_2O_2$: 3 $H_2SO_4$) for 20 min. Caution: piranha should be handled with extreme care. The clean electrode was then rinsed with MilliQ water and etched in degassed 40% NH$_4$F for 20 min. After drying under argon, the sample was placed in a Petri dish containing a neat sample of the alkyne molecule (either 1,8-nonadiyne or 1-nonyne, Fig. 1) and transferred to a custom-made ultraviolet reactor, and the reaction was allowed to take place overnight under an inert atmosphere. Upon completion of the hydrosilylation reaction, the silicon sample was rinsed thoroughly with ethanol before being analysed.

**Atomic force microscopy.** AFM imaging were carried out using a PicoSPM1 microscope (Molecular Imaging, Phoenix, AZ) in tapping mode with silicon nitride cantilevers (Nanoprobe, spring constant of 20 Nm$^{-1}$) in ambient conditions.

**Scanning tunnelling microscopy.** Single-molecule measurements were performed using an STM in a so-called blinking experiment. In the blinking BJ-STM approach, under a fixed Si-Au electrodes gap separation the STM tunnel-current feedback is turned off and once the system is stabilized, the current transients are captured. When a molecule is attached between both electrodes, a sudden jump or 'blink' in the captured current is obtained in the form of telegraphic blinks[24,42,43,53]. Conductance 1D histograms and 2D maps were built by the accumulation of hundreds of individual blinks for each of the studied systems. The blinking maps give average information about the conductivity of the

single-molecule junctions studied and their mechanical stability. The presented experiments were performed with a PicoSPM I microscope head controlled by a 'Picoscan 2500' electronics, from Agilent. The STM head was electronically and mechanically isolated. Data were acquired using a NI-DAQmx/BNC-2110 National Instruments (LabVIEW data acquisition System) and analysed with a code based on LabVIEW software.

**X-ray photoelectron spectroscopy.** XPS measurements were carried out in a PHI 5500 Multitechnique System (Physical Electronics) along with a monochromatic X-ray source (aluminium Kα line of 1486.6 eV energy and 350 W), placed in a perpendicular position to the analyzer axis and calibrated using the 3d5/2 line of Ag with a full width at half maximum of 0.8 eV. The area analysed was a 0.8 mm diameter circle. The selected resolution for the general spectra is 187.85 eV of pass energy and 0.8 eV per step. The selected resolution for the different elements is 23.5 eV of pass energy and 0.1 eV per step. The experiments were performed in an ultra-high vacuum chamber under a pressure between $5 \times 10^{-9}$ and $2 \times 10^{-8}$ torr.

**Materials.** All chemicals are of analytical grade and are not pre-treated before usage. Reagents used in the modification of surfaces and in the electrochemical measurements are of high purity (>99%). Single-side polished silicon wafers (Prime-grade), 111-oriented (<111> ± 0.5°), n-type (phosphorous-doped), 500 ± 25 μm thick, with resistivity 0.001–0.003 Ω cm resistivity were purchased from Siltronix, S.A.S. (Archamps, France) and are denoted as highly doped (Si$_{HD}$). Single-side polished silicon wafers, 111-oriented (<111> ± 0.09°), n-type (phosphorous-doped), 500 ± 25 μm thick, 1–10 Ω cm resistivity were purchased from Virginia Semiconductors, Inc. (Fredericksburg, VA, USA) and are denoted as low doping (Si$_{LD}$).

**Data availability.** Details of the experimental methods are available within the article and its Supplementary Information file, or available from the corresponding authors on request.

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

## Acknowledgements

This research is supported by the Spanish National Projects-MINECO (no. CTQ2012-36090, CTQ2015-71406-ERC and CTQ2015-64579-C3-3-P). N.D. thanks the Australian Research Council for a DECRA fellowship (DE160101101). I.D.-P. thanks the Ramon y Cajal programme (MINECO, no. RYC-2011-07951) for the financial support. S.C. thanks the Australian Research Council for a DECRA fellowship (DE160100732). A.C.A. thanks the Spanish Ministerio de Educación for an FPU fellowship. J.J.G. acknowledges the generous financial support from the Australian Research Council for an Australian Laureate Fellowship (FL150100060).

## Author contributions

N.D. and I.D.-P. conceived the idea. A.C.A. and N.D. performed the experiments and analysed the data. All authors contributed to the interpretation of the results. N.D. and I.D.-P. wrote the manuscript with contributions from all authors.

## Additional information

**Competing interests:** The authors declare no competing financial interests.

**Publisher's note**: 

