## [Peer Review File · Nature Communications]

Reviewers' comments:

Reviewer #1 (Remarks to the Author):

The paper describes the experimental demonstration of single-molecule diodes on silicon (111) substrates using STM-based break-junction techniques. I-V characteristics were measured during a "blinking" event in which the STM Au tip and the nonadiyne monolayer are connected on and off. It is shown that I-V curves show rectifying behavior when low-doped silicon is used, which is qualitatively explained in the framework of the standard MIS tunneling diode. The STM experiment is carefully performed in a systematic way as has been similar in previous reports from the authors' group, which makes it possible for them to access to the single molecule and provide clear data. However, while the paper should be published in suitable journals, my opinion is that it does not provide impactful findings required for Nature Communications as I explain below. The impact should be clarified in the following two aspects: 1) there have been several uncited reports on the measurement of single molecules on silicon such as Guisinger et al. (PNAS 2005) and Piva et al. (Nature 2005). The authors should discuss more on the novelty of the present work. 2) I think that this might be more crucial; the rectification ratio of about 4000 would not be very surprising in terms of MIS tunneling diodes using low-doped silicon. It is not due the single molecule and could be achieved with ensembles of molecules or without any molecules. Therefore it is not clear how the obtained ratio is impactful in the context of future application of molecular electronics. The author should have rather demonstrated how the accessibility to the single molecule could unveil new characteristics/information of molecules. One of the ways might be to analyze I-V curves more quantitatively to extract useful information. For example, Salmon (PRL 2005) has already reported on quantitative investigation of the electrical transport in metal-alkyl monolayers-Si junctions though it was on ensemble measurements.

I would like to add a few comments:

- 1) It would be natural to imagine that more than one molecule can bridge the gap from the statistical viewpoint. The author should discuss about it. It seems in Fig. 5(a) that the distribution of conductance has a few major branches. The author should discuss the possible origins.
- 2) As noted above, I wonder if the author could add more quantitative discussion and/or analysis on rectifying I-V curves; there is some discussion, but it is not very clear, lacking information of the energy gap between LUMO and HOMO and work-function difference. The band diagram in the inset of Fig. 4 seems too schematic. Furthermore, the units of the bias voltages, the level of conductance are inconsistent although these are important values to understand I-V curves.

Reviewer #3 (Remarks to the Author):

The paper by Albert C. Aragonès et al. presents a method to wire a symmetrical saturated alkyl chain molecule terminated with alkyne groups between gold and n-type silicon electrodes by the scanning tunneling microscopy break-junction technique. The silicon-based single-molecule diode shows high rectification ratios closing to 4000 when the low doped silicon wafers were used. This idea is interesting and the performance obtained in this work seems quite promising, however, there are still some major problems concerning the measurement should be discussed in more details. I will recommend several clarifications before further consideration that are listed below:

1. The single-molecule junction conductance may change considerably as the molecular configuration varies. What will be the molecular configuration in the measurement? A tilt one or fully stretched one? More details about the interrelation between conductance and molecular configuration should be

discussed.

2. The author claimed that the rectification ratios of gold/1,8-nonadiyne/silicon junction diode could exceeds 4000. However, control experiments should be conducted in order to exclude the intrinsic rectification ratios between the metal tip and semiconductor substrate without the molecules. I would suggest to testify the proportion of rectification ratios contributed from the contact of gold tip and silicon substrate.

3. Concerning charging effect, as shown in Figure 2 and Figure S1.1, when the 2-vertex working bias was ramped between +2 and -2 V from negative to positive potential, the low doped n-type silicon substrate electrodes shows very high current rectification, the same with Figure 4. How about changing the direction of the scanning voltage? Some more data are needed to exclude the influence from charging effect.

To sum up, I think this story present an interesting idea, and the work are carefully done. I would suggest a major revision for this paper, and I think the paper can be a quite nice paper after solving the above questions.

Reviewer #1 (Remarks to the Author):

The authors added new data and references to shift their main message from the high rectification ratio to the improved stability (reproducibility) in I-V curves achieved by the molecular bridge at ambient conditions that differ from UHV circumstances in the previous reports on single molecule junctions using STM. The authors now clearly state that the obtained high rectification is not caused by the molecule itself, but it is due to the MIS diode characteristics. Although the use of AFM tip even at ambient conditions seems still far from practical device application, I would appreciate these modifications since such stability just with a single-molecular junction would be a useful finding for the research community working on single molecule device. However, I feel that the way how the new data are added and shown looks somewhat intentional to support the story of the paper. Furthermore, the presentation style is not good enough in terms of scientific papers. Therefore I would like to request the authors to consider or show the data and their statistics more carefully to provide useful information, which I think is necessary for the publication in Nature Communications.

- 1) I-V curves are shown in linear plots. The authors should also show semi-log plots to give information on small current level in either the main body or SI.
- 2) In Fig. 2 the set voltage is about -0.4 V and the current goes up to -10 nA after the formation of the molecular bridge. In Fig. 4 (a), however, it looks that the voltage less than -1.0V is necessary to generate -10nA current. Is this discrepancy due to the instability of the Au-nonadiyne-Si system? I understand that the experiment is rather hard and very sophisticated, but once the authors discuss the stability/reproducibility, the authors should make efforts to explain all the related matters.
- 2) In the statistical comparison between Fig. 4 (a) and (b), the number of curves are rather different (23 to 33 curves), which looks unfair since 33 curves naturally show a large distribution.
- 3) The variation is not very different between Au- nonadiyne -Si (LD) and Au-Si(LD), which rather contradicts with the authors' claim.
- 4) The set voltage is -0.8 V in Fig. 5(b), which is not the same with -0.4 V used for Fig.2. Why is it? In Fig. 5 (b) how large is the current level after the breakdown? Is it not -1 nA, which is the set current? I suggest that the authors add the right axis to shows the current level. This is also the case for Fig. 6. G_0 will be the quantum conductance, but I could not find the definition in the paper.
- 5) The sentence of "The slight dispersion in conductance values observed in Fig. 5a might be due to intrinsic junction-to-junction variation in the details of the molecule/Au electric coupling, which could be amplified by the digital limitation of our analogic/digital signal converter or, in some instances, to the sudden formation of multi-molecular bridges." is not very clear to me due to the lack of quantitative discussion. How much is the resolution limit due to the AD converter? I wonder if it is really a partial reason for the dispersion. The author claimed that the dispersion is slight, but the difference in the conductance is rather large up to almost double the lowest branch. The author also noted that multi-molecular bridges can form for some instances. I would like to ask the authors to explain how they can identify the multi-molecular bridge or the variation of the single junction.
- 6) The variation is much smaller with gold substrate in Fig. S2. 1 compared to Fig. 5 (a) and Fig. S1. 3(a). Can Gold substrates be better in the conductance variation or is this due to the different current level? What will happen if the comparison is made in the same current level?
- 7) Fig. 6 (a) seems to show that the junction is easily formed in a time shorter than 0.5 s, which is inconsistent with 20 junction/hr noted in the text.
- 8) Why is the ohmic behavior observed in Fig. S1. 2 (c) ?
- 9) The authors may discuss if the formation/ breakdown is a Poisson process.
- 10) In Fig. 2 and Fig. 6 the horizontal axis is labelled with "junction lifetime", but it should be "time"

Reviewer #3 (Remarks to the Author):

We are happy with the response 2 and 3 to our comments, however, it seems the author still have not provided sufficient information about the possible molecular configuration(our comment 1). I am not quite agree that saturated molecule has no conductance changes upon tilting. Some more complementary reference and discussion are still suggested to clarify this point.

To sum up, I think the substantial additions on original submission of the manuscript makes the story much more convincing. After this minor change, I think the manuscript could be published.

Reviewer #4 (Remarks to the Author):

The manuscript entitled "Single-Molecule Electrical Contacts on Silicon Electrodes under Ambient Conditions" by Aragonés et al. describes the development of a Metal-Insulator-Semiconductor rectifier that uses a single molecule for the Insulator component. While such systems have been studied in large area devices, the push toward single-molecule devices from these structures is novel, and the observations of stability and lifetime add to this novelty. The work is interesting and timely as the push toward different electrode materials seems to be an emerging and necessary area of study for molecular electronics. The experimental aspects of the work are of high quality, and it is my opinion that this work will be of interest to the readers of Nature Communications. There are however, a few points that should be "rectified".

First, I don't understand the invocation of the A/D converter response to describe the variation of conductance values in Fig. 5. Given a current range of +/-100nA for their current amplifier (as seen in the I-V sweeps), even a 12-bit A/D should give a resolution of a few pA. Also, I'd suggest, developing a 1-D histogram from this data to see if multiple peaks are showing up to support the alternative explanation.

Second, in the discussion of the energy band profile in the semiconductor the authors state that they used a Poisson distribution in the semiconductor. This is unclear. Perhaps they mean a linear electric field profile coupled with Poisson's equation, but if they really mean a Poisson distribution, this discussion needs to be expanded.

Third, the discussion of switching frequency should be described in more detail. Intuitively, I would expect the opposite: the increased mobility would increase the switching probability, not decrease it.

Finally, changes in dispersion are typically quantified by producing conductance histograms. Although this information is contained in their time blink histograms, it is not as obvious in this presentation.

Reviewer #5 (Remarks to the Author):

I fully agree with the critical concerns by reviewer #1, especially those with the interpretation of the statistical analysis under varying conditions. For example, the different bias cannot justify the results and the corresponding conclusions. In summary, the paper should be rejected.

Reviewer #1 (Remarks to the Author):

The authors modified the manuscript properly in general and added statistical analysis to show the reproducibility of the sing-molecular junction on Si substrates. In the present manuscript, however, there seems to be still a few issues about precise description, which I think should be adequately addressed for the publication.

1) All the conditions used should be clearly described. It the initial set-point current -1 nA throughout the work? If not, please include all the information since some information might be missing due to the vertical offset used in the data plot; some figure options include the value for set-point current , but some do not. If the use of different set-point currents is necessary, is it due to instability of the molecular system or AFM ?

2) It is a little surprising to me that the authors stated that the previous Figure 2 was meant to be a schematic and they plotted -0.4 V wrong instead of the actual value -1.6 V. I would like to strongly request the authors to check if all the data and parameters are shown correctly. It is not clear to me why it was not possible to plot real voltages for the new data in Fig. 2 (b) and (c). If some of the data were lost, they should not be used.

Reviewer #3 (Remarks to the Author):

I am happy with the response to my comment in the second round. Although I personally still have slightly different opinion, I think the discussion for this point is now convincing.

I also went through the response to reviewer 1,4 and 5. Actually I think 3 reviewers among all 4 agreed that it is an interesting story that it present something beyond the current state-of-art in molecular electronic community. I have the feeling that the first submission seems slightly too rush from the authors, but the revised version looks much better. Some comments from reviewer 1 looks like from a very physical community, but I think the author addressed most of them well. The understanding of the measurment data in a novel system always required huge effort, and I think the authors have deal with enough care. There is always some argument in single-molecule electronics community that whether the statistics is careful enough or not, and I would say the approach author applied in this work should be above average on this point, and the statistics of data are good enough to support the conclusion.

To sum up, I suggest that the paper could be published.

Reviewer #4 (Remarks to the Author):

The authors have made significant improvements to the manuscript, I am satisfied with their changes, and believe this manuscript is suitable for Nature Communications.

Reviewer 1:

The paper describes the experimental demonstration of single-molecule diodes on silicon (111) substrates using STM-based break-junction techniques. I-V characteristics were measured during a "blinking" event in which the STM Au tip and the nonadiyne monolayer are connected on and off. It is shown that I-V curves show rectifying behavior when low-doped silicon is used, which is qualitatively explained in the framework of the standard MIS tunneling diode. The STM experiment is carefully performed in a systematic way as has been similar in previous reports from the authors' group, which makes it possible for them to access to the single molecule and provide clear data. However, while the paper should be published in suitable journals, my opinion is that it does not provide impactful findings required for Nature Communications as I explain below. The impact should be clarified in the following two aspects:

1) there have been several uncited reports on the measurement of single molecules on silicon such as Guisinger et al. (PNAS 2005) and Piva et al. (Nature 2005). The authors should discuss more on the novelty of the present work. 2) I think that this might be more crucial; the rectification ratio of about 4000 would not be very surprising in terms of MIS tunneling diodes using low-doped silicon. It is not due the single molecule and could be achieved with ensembles of molecules or without any molecules. Therefore it is not clear how the obtained ratio is impactful in the context of future application of molecular electronics. The author should have rather demonstrated how the accessibility to the single molecule could unveil new characteristics/information of molecules. One of the ways might be to analyze I-V curves more quantitatively to extract useful information. For example, Salmon (PRL 2005) has already reported on quantitative investigation of the electrical transport in metal-alkyl monolayers-Si junctions though it was on ensemble measurements.

Response: We have revised the manuscript according to the reviewer's suggestion and added new data and text to specifically highlight on the conceptual difference between a single-molecule junction and a macroscopic molecular junction on a semiconductor. We agree with the reviewer that the previous version of the manuscript appeared to have a strong focus on the magnitude of the current rectification. Trapping a single chemical entity is a necessary step to access reproducible charge transport data under ambient conditions. We believe it was not explicitly clear in the submitted manuscript how a single molecule contact is key to reproducible measurements of current-voltage responses at room temperature and atmospheric pressure.

We note that the articles referred to by the Ref #1 (Guisinger *et al.*¹ and Piva *et al.*², now cited in the revised manuscript) did focus on charge transport STM measurements in either cryogenic ultra-high vacuum or ultra-high vacuum set ups. To date, low temperature and ultra-high vacuum have been essential to minimize the oxidation, chemical contamination of the silicon surface, the thermal drift and piezoelectrical creep. Furthermore, the STM set ups of previously published work are "contact-less", *i.e.* all these experiments are performed in the absence of a chemical contact between the STM tip and the Silicon substrate. The lack of a chemical contact with both top and bottom electrode does inevitably lead to a data scatter or artefacts in the current-voltage measurements. This is due to slight change in the tip-molecule spacing inevitably resulting in changes to the fraction of the total voltage drop across the molecular junction. This becomes a serious issue for the measurement of exponentially decaying tunnelling currents as in molecular electronics type devices. Even more significant, when these measurements start to be performed under ambient conditions, which is a prerequisite for any molecular electronics device application, a chemical "lock" of the distance between contacts becomes a must and we have shown how to achieve and probe that this contact is in place.

The second issue the reviewer raises is the comparison with previous studies on ensemble of molecules or large-scale molecular junctions. We note that large-scale molecular junctions suffer

from technical drawbacks as a fundamental platform to study chemistry-related charge transport. For instance, it is known that direct deposition of gold contacts onto molecular monolayers covalently bonded to silicon surfaces causes penetration of the metal into the monolayer (pinholes), resulting in deteriorated performance and poor reproducibility.^{3,4} There are also issues regarding the quality of the contact between the top electrode and the organic layer, which affects current responses. For example, current-voltage properties of a large-scale tunnelling junctions can range from those of a resistor to those of a diode depending on the size and geometry of the top-metal contact (see Fig. R1).^{5,6} Hence, another approach is needed to control the chemical and physical properties of the top-contact between the metal electrode and the monolayer-covered silicon electrode.

Figure R1: from reference 6: the current density of the junction varies from that of a diode to that of a resistor depending on the geometry of the contact.

We argue that covalently bonding single-molecules between a metal and silicon electrodes solve most of the issues of large-scale molecular junctions thus accessing the fine details of the semiconductor/molecule/metal junction: it controls the distance between the electrodes (dictated by the length of the molecule), it limits the variation in the contact geometry and preserve the chemical purity between the junction electrodes as we show below in our new set of control experiments, now included in the revised version of the manuscript. More importantly, access to single molecules allows measurements to be performed at ambient conditions without the need of ultrahigh vacuum or cryogenic temperature controls, issues that would significantly limit the application of these molecular scale electrical contacts.

Changes made in the revised manuscript:

In the revised manuscript, we presented new experiments that explicitly show the importance of a single-molecule contact on silicon electrodes to control the current rectification ratio under ambient conditions. This is now included in the new section of the manuscript: “Characterizing the electrical properties of the junction in the absence of molecules”. As a consequence, we have made a slight modification in the title to “*Single-Molecule Electrical Contacts on Silicon Electrodes under Ambient Conditions*” in order to better reflect on the changes to the text and Figures.

We have collected I-V curves in the absence of the nonadiyne molecules (see Fig. R2- R2a panel is now in Fig. 4b of the manuscript and Fig. R2 has been added to the SI section 1). Hence the voltage

was ramped at either a fixed position above a H-terminated silicon or at different X,Y or Z positions. In the absence of a molecule between the semiconductor and the metal contact, a consistent current response cannot be obtained due to either variation in the tunnelling distance (Z direction) that largely oscillate at room temperature, or a possible variation in the chemical functional groups on the silicon substrate along the XY direction (most likely surface dipoles from the SiOx contaminants). Unlike the case of a single-molecule junction with nonadiyne, the “empty” tunnelling junctions showed an unpredictable rectification ratios regardless of the doping level of the underneath silicon substrate (Fig. R2).

Figure R2. I(V)s characteristics of (a) a Au-Si_{HD} tunnelling gap, (b) a Au-Si_{LD} tunnelling gap and (c) a Au/Si_{LD} in direct physical contact showing an Ohmic behaviour. The non-rectifying and rectifying behaviours can be observed on both (a) and (b) gap junctions with a large dispersion in the RR values, showing the poor reproducibility of an open tunnelling gap under ambient conditions.

The effect of the distance between the junction electrodes becomes even more critical when the Au STM tip physically contacts the silicon substrate, which changes the I-V properties to that of an Ohmic contact (symmetrical, linear I-V) regardless of the doping level used in the silicon (see Fig. R2c).

In addition to the unique reproducibility in the electrical properties of single-molecule contacts on silicon, we further emphasize in the revised manuscript that the junctions performed on silicon electrodes have superior mechanical properties to that performed on the typical gold electrodes. This has implications in one of the major challenges in molecular electronics, which is the search

for a stable and robust (non-labile) contact.⁷ We found that the frequency of the junctions increases 2.5 folds (20 junction/hr to 50 junctions/hr) when a silicon substrate was used as the base electrode (see Fig. R3, now Fig. 6 in the manuscript). This enhanced junction formation is attributed to the absence of mobility of the silicon atoms as opposed to that high mobility of gold atoms in gold electrodes.

Figure R3. Blinking experiments of 1,8-nonadiyne on different substrates. Representative individual current traces showing blinking events corresponding to the spontaneous formation of single-molecule junctions of 1,8-nonadiyne molecules on Si (a) and gold (b) substrates. The applied surface-voltage bias were -800 mV and -200 mV for Si and Au substrates respectively, and 1 nA initial setpoint current in both cases.

We have also modified the text in page 1-2 to emphasise on the necessity of a single molecule contact:

Page 1: “Most efforts to date to prepare *metal/molecule/silicon* electrical contacts have involved macroscopic devices (ca. 10^{-3} cm²) comprising very large ensembles of molecules.⁸⁻¹⁰ The top contact with a second electrode is usually made via metal deposition on the top of the organic layer either by evaporation or *via* Hg or alloy-metal contacts.^{11,12} This macroscopic approach has provided most of the knowledge we have about charge transport across the molecule-semiconductor junction, however, they do suffer from critical drawbacks as fundamental platforms for charge transport studies. For instance, it is known that direct deposition of gold contacts onto molecular monolayers covalently bonded to silicon surfaces causes penetration of the metal into the monolayer (pinholes) resulting in deteriorated performance and poor reproducibility.^{3,4} There are also issues regarding the quality of the contact between the top electrode and the organic layer that has enormous effects on the current responses. For instance, current-voltage properties of a large-scale tunnelling junctions can range from those of a resistor to those of a diode depending on the size and geometry of the top-metal contact.^{5,6} Hence, another approach is needed to control the chemical and physical properties of the top-contact between the metal electrode and the monolayer-covered silicon electrode. One of the solutions would be a contact between the top-metal electrode and an individual molecule on the silicon substrate, which allows for a fixed distance between the junction electrodes and would limit to one (the molecular backbone) the channels across which electrons transport.”

Page 2: “We show that we can control under ambient conditions the tunneling distance and remove the intrinsic limitations of charge transport STM measurements in ultra-high vacuum and/or cryogenic setups.^{1,2} This approach also offers a fundamental platform to study molecular charge transport addressing the variability issue of large area metal-organic insulator-semiconductor measurements.

Reviewer 1:

would like to add a few comments:

1) It would be natural to imagine that more than one molecule can bridge the gap from the statistical viewpoint. The author should discuss about it. It seems in Fig. 5(a) that the distribution of conductance has a few major branches. The author should discuss the possible origins.

2) As noted above, I wonder if the author could add more quantitative discussion and/or analysis on rectifying I-V curves; there is some discussion, but it is not very clear, lacking information of the energy gap between LUMO and HOMO and work-function difference. The band diagram in the inset of Fig. 4 seems too schematic. Furthermore, the units of the bias voltages, the level of conductance are inconsistent although these are important values to understand I-V curves.

Response:

1) We agree with the reviewer that there exists a slight variation in the *blinking* conductance values. We ascribed the conductance dispersion mainly to different details of the molecule/Au contact yielding slight electrical coupling differences. Multi-molecular junctions are usually observed as a set of conductance values with integer values of the single-molecule conductance and their probability appears to be largely diminished¹³. The time to hold two molecules in the junction at the same time is then usually below the time resolution of our current amplifiers and then rarely observed in a *blinking* experiment. We have added a comment to clarify this point: "The slight dispersion in conductance values observed in Fig. 5a might be due to intrinsic junction-to-junction variation in the details of the molecule/Au electric coupling, which could be amplified by the digital limitation of our analogic/digital signal converter or, in some instances, to the sudden formation of multi-molecular bridges". Moreover, we note that these variations are significantly less than that obtained with the tunnelling junctions in the absence of molecules or in large-scale molecular junctions, where a top contact with billions of molecules can significantly vary from junction to junction as shown in our response to the reviewer's first question.

We note that all our measured I-V curves were performed while a single-molecule was still connected to both junction electrodes.⁶ This was confirmed by monitoring the current after the bias ramp, which was the same as before the bias ramp. We then performed mechanical pulling to check whether there exist other current plateaus (molecular junctions) as the tip was pulled away from the junction (Fig. 3 of the manuscript).

2) We have extended Figure 4 of the manuscript (see Figure R4, now panel c in Figure 4 of the manuscript) to re-sketch the band diagrams and introduced details on the energetics and the different transport regimes along the I(V) characteristic. We have represented the HOMO-LUMO levels above and below the formal Si band gap since the HOMO-LUMO gap is much larger for a saturated alkane molecule (~8.07 eV) from our gas-phase DFT calculations that have now been added to the SI.

We thank the reviewer to point the inconsistency in the voltage units (mV vs V) and we have now fixed all the typos of the voltage units in the revised manuscript.

Figure R4. Energy band diagrams representations of the Au-nonadiyne-Si junction under charge carriers' accumulation conditions (applied negative sample voltages) in the Si electrode (left panel) and under depletion conditions (applied positive sample voltage) for the Si_{HD} (central panel) and Si_{LD} (right panel).

Reviewer #3 (Remarks to the Author):

The paper by Albert C. Aragonès et al. presents a method to wire a symmetrical saturated alkyl chain molecule terminated with alkyne groups between gold and n-type silicon electrodes by the scanning tunneling microscopy break-junction technique. The silicon-based single-molecule diode shows high rectification ratios closing to 4000 when the low doped silicon wafers were used. This idea is interesting and the performance obtained in this work seems quite promising, however, there are still some major problems concerning the measurement should be discussed in more details. I will recommend several clarifications before further consideration that are listed below:

1. The single-molecule junction conductance may change considerably as the molecular configuration varies. What will be the molecular configuration in the measurement? A tilt one or fully stretched one? More details about the interrelation between conductance and molecular configuration should be discussed.

We thank the reviewer to point out this issue. We have originally considered this particular issue and for this reason we have used a completely saturated molecule that lacks pi-orbitals in its backbone and, hence, has no effects on the conductivity measurements upon tilting.¹⁴ We have also used the blinking, which allows the spontaneous formation of molecular junctions and therefore minimizes the variation in molecular contact geometry since we usually observe the most stable ones. We note however that statistically there exist some variation in the current response, though limited, that could originate from different contact configurations at the

electrodes/molecule interface or different stretching degrees of the molecular bridge during the single-molecule experiment. We have now included in the revised manuscript the following to clarify this point: “The slight dispersion in conductance values observed in Fig. 5a might be due to intrinsic junction-to-junction variation in the details of the molecule/Au electric coupling, which could be amplified by the digital limitation of our analogic/digital signal converter or, in some instances, to the sudden formation of multi-molecular bridges.”

2- The author claimed that the rectification ratios of gold/1,8-nonadiyne/silicon junction diode could exceeds 4000. However, control experiments should be conducted in order to exclude the intrinsic rectification ratios between the metal tip and semiconductor substrate without the molecules. I would suggest to testify the proportion of rectification ratios contributed from the contact of gold tip and silicon substrate.

We have performed these experiments according to both reviewers' suggestions and introduced the changes pointed in the reviewer 1's answer. As explained in our response to reviewer 1, our additional control experiments show that the current rectification ratio depends extremely on the distance between the electrodes, which can be controlled by the formation of the molecular bridge. In the extreme case, when a gold tip is in contact with a silicon electrode (in the absence of molecules), current rectification disappears and the contact becomes Ohmic (see Fig. R5c, now included in SI section 1). The current rectification is regained to a certain degree when moving the gold tip away from the silicon surface; however, the I-V response ranges from that of a resistor (weak or no current rectification) to that of a diode randomly as the tip drift in the X, Y or Z direction (see Fig. R3a-b). Moreover, the presence of the molecular layer on the Si electrode prevents its surface oxidation, which leads to another source of variability in the electrical properties of the Si/Au interface.

Figure R5. $I(V)$ s characteristics of (a) a Au-Si_{HD} tunnelling gap, (b) a Au-Si_{LD} tunnelling gap and (c) a Au/Si_{LD} in direct physical contact showing an Ohmic behaviour. The non-rectifying and rectifying behaviours can be observed on both (a) and (b) gap junctions with a large dispersion in the RR values, showing the poor reproducibility of an open tunnelling gap under ambient conditions.

These control experiments show that single-molecule contacts are necessary to control the electrical properties of the junction. First, it prevents the metallic tip from physically contacting the silicon surface, which will then produce a non-rectifying Ohmic contact, and they also prevent the H-terminated from oxidation which would also change the current response.

3-Concerning charging effect, as shown in Figure 2 and Figure S1.1, when the 2-vertex working bias was ramped between +2 and -2 V from negative to positive potential, the low doped n-type silicon substrate electrodes shows very high current rectification, the same with Figure 4. How about changing the direction of the scanning voltage? Some more data are needed to exclude the influence from charging effect

The IV curves were performed in both sweep directions and we observe no hysteresis. We have now clarified this point by adding a new figure in the SI section 1 showing an $I(V)$ curve with both forward and backward scans.

References for this letter:

1. Guisinger, N. P., Yoder, N. L. & Hersam, M. C. Probing charge transport at the single-molecule level on silicon by using cryogenic ultra-high vacuum scanning tunneling microscopy. *Proc. Natl. Acad. Sci. U. S. A.* **102**, 8838–43 (2005).
2. Piva, P. G. *et al.* Field regulation of single-molecule conductivity by a charged surface atom. *Nature* **435**, 658–661 (2005).
3. Reus, W. F. *et al.* Statistical Tools for Analyzing Measurements of Charge Transport. *J. Phys. Chem. C* **116**, 6714–6733 (2012).
4. Zhang, Z.-S. & Rabson, D. A. Electrical and thermal modeling of the non-Ohmic differential conductance in a tunnel junction containing a pinhole. *J. Appl. Phys.* **95**, 557 (2004).
5. Ru, J., Szeto, B., Bonifas, A. & McCreery, R. L. Microfabrication and Integration of Diazonium-Based Aromatic Molecular Junctions. *ACS Appl. Mater. Interfaces* **2**, 3693–3701 (2010).
6. Zhu, L., Popoff, R. T. W. & Yu, H.-Z. Metastable Molecular Metal–Semiconductor Junctions. *J. Phys. Chem. C* **119**, 1826–1831 (2015).
7. Widawsky, J. R. *et al.* Length-Dependent Thermopower of Highly Conducting Au–C Bonded Single Molecule Junctions. *Nano Lett.* **13**, 2889–2894 (2013).
8. Yuan, L., Breuer, R., Jiang, L., Schmittel, M. & Nijhuis, C. A. A Molecular Diode with a Statistically Robust Rectification Ratio of Three Orders of Magnitude. *Nano Lett.* **15**, 5506–12 (2015).
9. Yaffe, O. *et al.* Molecular electronics at metal/semiconductor junctions. Si inversion by sub-nanometer molecular films. *Nano Lett.* **9**, 2390–4 (2009).
10. Vilan, A. *et al.* Molecules on si: electronics with chemistry. *Adv. Mater.* **22**, 140–59 (2010).
11. Haick, H. & Cahen, D. Making contact: Connecting molecules electrically to the macroscopic world. *Progress in Surface Science* **83**, 217–261 (2008).
12. Popoff, R. T. W., Kavanagh, K. L. & Yu, H.-Z. Preparation of ideal molecular junctions: depositing non-invasive gold contacts on molecularly modified silicon. *Nanoscale* **3**, 1434–45 (2011).
13. Xu, B. & Tao, N. J. Measurement of single-molecule resistance by repeated formation of molecular junctions. *Science* **301**, 1221–3 (2003).
14. Diez-Perez, I. *et al.* Controlling single-molecule conductance through lateral coupling of π orbitals. *Nat. Nanotechnol.* **6**, 226–31 (2011).

Reviewer # 1

The authors added new data and references to shift their main message from the high rectification ratio to the improved stability (reproducibility) in I-V curves achieved by the molecular bridge at ambient conditions that differ from UHV circumstances in the previous reports on single molecule junctions using STM. The authors now clearly state that the obtained high rectification is not caused by the molecule itself, but it is due to the MIS diode characteristics. Although the use of AFM tip even at ambient conditions seems still far from practical device application, I would appreciate these modifications since such stability just with a single-molecular junction would be a useful finding for the research community working on single molecule device. However, I feel that the way how the new data are added and shown looks somewhat intentional to support the story of the paper. Furthermore, the presentation style is not good enough in terms of scientific papers. Therefore I would like to request the authors to consider or show the data and their statistics more carefully to provide useful information, which I think is necessary for the publication in Nature Communications.

We agree with Reviewer 1 that the presentation of our data in few places in the old version of the manuscript was crude and warrants clarification; however we stress that there is nothing intentional or misleading. As suggested by Reviewer 1, we have expanded, both in the text as well as in Figures (Fig. 4a and 4b as well as Fig.S1.2a-b) on the statistical analysis of the data. We have made every effort to present our data in a clearer way and to clarify points that could be a source of confusion. Sampling sizes, e.g. when comparing single-molecule junctions versus the empty junctions, have been matched and relative coefficient of variation (C_V) has been added to the discussion (see answer to question 4). We have also answered and clarified all of Reviewers' specific inquiries and modified the text and figures (new figures S1, S1.1.3, S1.1.4, S1.2d and S1.4.2) as detailed below:

1-I-V curves are shown in linear plots. The authors should also show semi-log plots to give information on small current level in either the main body or SI.

Our single-molecule experimental setup, like many other similar STM-based instruments (e.g. *Nat. Nanotechnol.* **7**, 35–40 (2011), *Nat. Mater.* **5**, 995–1002 (2006) is based on linear current amplification, hence log-plots within the current range accessible do not add clarity to the presentation of data. IV curves of single molecule have also been largely reported in the linear scale (e.g. *Nat. Comm.* **6**, 6324 (2015), *Nat. Nanotechnol.* **9**, 830–834 (2014), *Nano Lett.* **13**, 6233–6237 (2013)). In the revised manuscript, we provide the coefficient of variation (C_V) for each system (Table 1.3, SI page 3). (see answer to question 4).

2- In Fig. 2 the set voltage is about -0.4 V and the current goes up to -10 nA after the formation of the molecular bridge. In Fig. 4 (a), however, it looks that the voltage less than -1.0V is necessary to generate -10nA current. Is this discrepancy due to the instability of the Au-nonadiyne-Si system? I understand that the experiment is rather hard and very sophisticated, but once the authors discuss the stability/reproducibility, the authors should make efforts to explain all the related matters.

We agree with the reviewer. Figure 2 was meant to be a schematic to explain the experiments and the bias set in previous version of Figure 2b,c was -1.6 V and is a bias higher than that used to collect the IV curves (-0.8 V). The higher bias was chosen to magnify the gap between the blink current signal and the baseline in figures 2b,c. We stress that the actual working bias for all IV-curves was set to -0.8 V and not to -0.4 V. We have now changed this in all figure 2 such that the bias used is equivalent to the bias (-0.8 V) used in the rest of the manuscript (IV collection). We have also set the bias sweep schematic panels in Figure 2 (bottom panels) to -0.8 V and noted that these panels are schematic only and are not an experimental output.

We have added Figure S1.1.3 in the SI showing that IV on Si_{HD} and Si_{LD} starts and ends at -0.8 V. We also show blinks of Si_{HD} in the SI (Figure S1) performed at different biases (including the -1.6 V of the older version of Fig 2b) with the aim to show examples of the previous “screening” blinking experiments performed before the selection of -0.8 V as the working bias. We have also added Figure S1.1.4 that shows that the IV curves do not change when the working bias (initial starting and ending bias) is changed and referred to it in the caption of Figure 4: “Changing the starting bias did not affect the shape of the IV curves (Supplementary Fig.S1.1.4) “

3- In the statistical comparison between Fig. 4 (a) and (b), the number of curves are rather different (23 to 33 curves), which looks unfair since 33 curves naturally show a large distribution.

Indeed a comparison would be more straightforward if made between equivalent numbers of samples. We have removed any ambiguity about statistical analysis by increasing the size of the sample to 35 measurements for both the single-molecule junction and the empty-junctions control experiments. We have now modified this in Figure 4a,b and S1.2.1a,b. We note that it is evident that in the absence of a molecule the coefficient of variation (C_v) for the RR values increases 6 folds (for Si_{HD} case) and 8 folds for the (Si_{LD} case) (Table 1.3, SI page 3).

The coefficient of variation (C_v) was obtained using the following expression:

$$C_v = \frac{\sigma}{\mu}$$

Where σ is the standard deviation and μ the mean.

C _v (Coefficient of variation)	Au-nonadiyne-Si _{LD}	Au-nonadiyne-Si _{HD}	Au-Si _{HD}	Au-Si _{LD}
	0.74	1.39	8.40	5.94

4- The variation is not very different between Au- nonadiyne –Si (LD) and Au-Si(LD), which rather contradicts with the authors’ claim.

To eliminate any ambiguity or underestimation of the difference between Au- nonadiyne -Si (LD) and Au-Si(LD) from the figures, we now have included numerical comparison between the coefficient of variation (C_v) of these two samples which clearly highlight the difference. In the absence of a molecule, the C_v increases 8 folds for the Si_{LD} case and 6 folds for the Si_{HD} (Table S1.3, SI page 3).

5- The set voltage is -0.8 V in Fig. 5(b), which is not the same with -0.4 V used for Fig.2. Why is it? In Fig. 5 (b) how large is the current level after the breakdown? Is it not -1 nA, which is the set current? I suggest that the authors add the right axis to shows the current level. This is also the case for Fig. 6. G₀ will be the quantum conductance, but I could not find the definition in the paper.

As mentioned in the reply to question 2, the working bias is -0.8 V and this has now been corrected in the revised manuscript. As requested, we have now added the axis on the right side of the graphs in figure 5 and 6 that shows the current level. We have also added the definition of G₀ in page 7 line 5. In Figure 5(b) there is no junction breakdown since the monolayer is formed from 1-nonyne which lack the distal alkyne moieties necessary for the formation of junctions. In the experiments of Figure 5a, the current drops down back to nearly 1nA (the setpoint current) after the junction breakdown. Both figures 5 a and b do not show the 1nA tunnelling current (background) because the background

is offset as stated in the figure caption “The 2D-maps were normalized versus the total amount of counts and the baseline has been vertically offset to zero-current to display the single-molecule junction component only (tunnelling background subtracted)”

6- The sentence of “The slight dispersion in conductance values observed in Fig. 5a might be due to intrinsic junction-to-junction variation in the details of the molecule/Au electric coupling, which could be amplified by the digital limitation of our analogic/digital signal converter or, in some instances, to the sudden formation of multi-molecular bridges.” is not very clear to me due to the lack of quantitative discussion. How much is the resolution limit due to the AD converter? I wonder if it is really a partial reason for the dispersion. The author claimed that the dispersion is slight, but the difference in the conductance is rather large up to almost double the lowest branch. The author also noted that multi-molecular bridges can form for some instances. I would like to ask the authors to explain how they can identify the multi-molecular bridge or the variation of the single junction.

We agree this point warrant a further discussion. The AD converter has a resolution limit of few pA and therefore is unlikely to contribute to the dispersion in conductance. Multi-molecular bridges normally show conductance values of multiple integers (Reference 2, *Science* **301**, 1221–3 (2003)). Hence, if a conductance band splits into two with the upper band having twice as much conductivity, this can be ascribed to a formation of double junctions. The fact that we don’t see a clear split of bands in our case (Figure 5), limits the capability in distinguishing between multi-molecular junction and variation in single molecule junctions. Another source of conductance variation can be attributed to the uncertainty in the stereochemistry of the silylated olefin (reaction of Si-H with 1-alkynes). The produced silylated olefin could be either *trans* or *cis* (*J. Am. Chem. Soc.* 2011, 133, 4998–5008) which could have different conductivities. The different bias applied (-0.8 V for silicon versus -0.2 V for gold to maintain the current for both cases in the ca. 3 nA range) can also induce more fluctuation in the silicon case, since the imposed electric field is much higher in this case and can lead to the stabilization of higher energy configurations of the molecular junction.

We have added this discussion for Figure 5 in line 4 page 7:

“The dispersion in conductance values (between $3.45 \cdot 10^{-5}$ and $7.00 \cdot 10^{-5} G_0$) observed in Fig. 5a might be due to intrinsic junction-to-junction variation in the details of the molecule/gold electrode contact geometry during stretching (*J. Am. Chem. Soc.* 128, 15874–15881 (2006) and *J. Am. Chem. Soc.* **128**, 2135–2141 (2006) or in some instances, to the formation of multi-molecular bridges. Other sources of conductance variation can be attributed to 1) the uncertainty in the stereochemistry of the silylated olefin (reaction of Si-H with 1-alkynes). The produced silylated olefin could be either *trans* or *cis* (*J. Am. Chem. Soc.* 2011, 133, 4998–5008) which could have different conductivities and 2) the different bias applied (-0.8 V for silicon versus -0.2 V for gold to maintain the current for both cases in the ca. 3 nA range) which can induce more fluctuation in the silicon case, since the imposed electric field is much higher in this case and can lead to the stabilization of higher energy configurations of the molecular junction.

7- The variation is much smaller with gold substrate in Fig. S2. 1 compared to Fig. 5 (a) and Fig. S1. 3(a). Can Gold substrates be better in the conductance variation or is this due to the different current level? What will happen if the comparison is made in the same current level?

We currently are not able to draw a clear conclusion of which is better, gold or silicon, regarding the dispersion in the conductance of single-molecule values. This is due to the presence of at least one gold electrode in both cases. As mentioned in our reply to question 6, there is also the possibility that the uncertainty in the stereochemistry of the silylated olefin (reaction of Si-H with 1-alkynes) is contributing to the conductance variation in the silicon case. The produced silylated olefin could be either *trans* or *cis* (J. Am. Chem. Soc. 2011, 133, 4998–5008), which could have different conductivities. Other Sources of variation can be attributed to the different bias applied (-0.8 V for silicon versus -0.2 V for gold to maintain the current for both cases in the ca. 3 nA range), which can induce more fluctuation in the silicon case. We note that the current levels in both the silicon and gold cases are comparable ($\sim 3\text{nA}$).

8- Fig. 6 (a) seems to show that the junction is easily formed in a time shorter than 0.5 s, which is inconsistent with 20 junction/hr noted in the text.

Figure 6 a and b are representative captures and do not resemble the average blinking frequency (50 blinks/hr) for the silicon case. In general, the x-axis therefore does not represent a continuous time axis and time-breaks between each blink should be added. We note however that with the silicon case, we occasionally observed more than one blink appearing in the 3 second frame (35% of the blinks in the silicon case appeared accompanied by other blinks within a 3 seconds timescale versus 5 % in the gold case). Figure 6a in the submitted manuscript represents a high-frequency blinking capture in a 3 second frame and hence was depicted without time breaks. We speculate that the non-homogenous and sudden increase in the frequency of the junction in the silicon case is attributed to the rigidity of the carbon-silicon bond which enables the same single molecule to form a junction multiple times in short period of time (breaking from the weak gold electrode and then reforming again) which lead to a sudden increase in the blinking frequency.

To avoid confusion, we have used other representative blinks for the silicon case and we added time breaks in the x-axis. We have moved the high frequency multi-blinks to the SI and described its occurrence in the text (Page 7 line 22):

“This is supported by occasionally observing more than one blink appearing in a 3 second frame (35 % of the blinks in the silicon case appeared accompanied by other blinks versus 5% in the gold case). Supplementary Fig. S1.4.2 shows an example of a high-frequency blinking capture in a 3 second frame. We speculate that the non-homogenous and sudden increase in the frequency of the junction in the silicon case is attributed to the rigidity of the carbon-silicon bond which enables the same single molecule to form a junction multiple times in short period of time (breaking from the weak gold electrode and then reforming again) which would lead to a sudden increase in the blinking frequency..”

9- Why is the ohmic behavior observed in Fig. S1. 2 (c) ?

We ascribe the Ohmic behaviour to gold leaking from gold tip to the silicon surface thus forming gold-gold-silicon junctions. To confirm this, we have now added to the SI experiments (Figure 1.2d) in which we deliberately deposit a ca. 25 nm thin film of gold on Si_{LD} . The I-V curves of this gold-gold- Si_{LD} interface are comparable to when an STM tip is in contact with Si_{LD} . Such interfaces have been reported to behave as Ohmic contacts and is attributed to surface states or recombination centres created by interfacial Au in Si (references 47-49).

We have added text page 6 line 43:

“This Ohmic behaviour is attributed to gold leaking from gold tip to the silicon surface thus forming gold-gold-silicon junctions. This is supported by a similar Ohmic behaviour recorded for I-V curves collected when the STM tip is in contact with a Si_{LD} surface covered by a thin film of gold (~25 nm). Such interfaces have been reported to behave as Ohmic contacts and is attributed to interfacial dipoles from high concentrations of interfacial charge or due to recombination centres created by interfacial Au in Si (References 47-49) of the revised manuscript).”

10- The authors may discuss if the formation/ breakdown is a Poisson process.

The formation/breakdown of the junctions is not Poisson process. For instance, we observed sudden increase in the frequency of the junctions and therefore the formation of the junctions appears to depend on the formation of the preceding junctions.

11- In Fig. 2 and Fig. 6 the horizontal axis is labelled with “junction lifetime”, but it should be “time”

We have now changed the term “junction lifetime” to “time”

Reviewer #3 (Remarks to the Author):

We are happy with the response 2 and 3 to our comments, however, it seems the author still have not provided sufficient information about the possible molecular configuration (our comment 1). I am not quite agree that saturated molecule has no conductance changes upon tilting. Some more complementary reference and discussion are still suggested to clarify this point.

To sum up, I think the substantial additions on original submission of the manuscript makes the story much more convincing. After this minor change, I think the manuscript could be published.

What we meant by “saturated molecules do not change their conductivity when tilting” is that the backbone of the molecule, upon tilting, does not contribute to the variation in the conductivity (*Nat. Nanotechnol.*, 2011, **6**, 226–31.). However, we do agree with the reviewer that even saturated molecules show variation in conductivity with tilting that is normally assigned to variation in the contact geometries between the molecule and the electrodes.

We have added this discussion for Figure 5 in line 4 page 7:

“The dispersion in conductance values (between $3.45 \cdot 10^{-5}$ and $7.00 \cdot 10^{-5}$ G_o) observed in Fig. 5a might be due to intrinsic junction-to-junction variation in the details of the molecule/gold electrode contact geometry during stretching (*J. Am. Chem. Soc.* 128, 15874–15881 (2006) and *J. Am. Chem. Soc.* **128**, 2135–2141 (2006)) or in some instances, to the formation of multi-molecular bridges. Other sources of conductance variation can be attributed to 1) the uncertainty in the stereochemistry of the silylated olefin (reaction of Si-H with 1-alkynes). The produced silylated olefin could be either *trans* or *cis* (*J. Am. Chem. Soc.* 2011, 133, 4998–5008) which could have different conductivities and 2) the different bias applied (-0.8V for silicon versus -0.2V for gold to maintain the current for both cases in the ca. 3 nA range) which can induce more fluctuation in the silicon case.

Reviewer #4 (Remarks to the Author):

The manuscript entitled “Single-Molecule Electrical Contacts on Silicon Electrodes under Ambient Conditions” by Aragonés et al. describes the development of a Metal-Insulator-Semiconductor rectifier that uses a single molecule for the Insulator component. While such systems have been studied in large area devices, the push toward single-molecule devices from these structures is novel, and the observations of

stability and lifetime add to this novelty. The work is interesting and timely as the push toward different electrode materials seems to be an emerging and necessary area of study for molecular electronics. The experimental aspects of the work are of high quality, and it is my opinion that this work will be of interest to the readers of Nature Communications. There are however, a few points that should be “rectified”.

First, I don't understand the invocation of the A/D converter response to describe the variation of conductance values in Fig. 5. Given a current range of +/-100nA for their current amplifier (as seen in the I-V sweeps), even a 12-bit A/D should give a resolution of a few pA. Also, I'd suggest, developing a 1-D histogram from this data to see if multiple peaks are showing up to support the alternative explanation.

We agree with the reviewer that the current amplifier is not the main source of variation in the conductance values. As mentioned in our reply to Reviewers 1 and 3, we have added a text discussing the source of variation in page 7:

The dispersion in conductance values (between $3.45 \cdot 10^{-5}$ and $7.00 \cdot 10^{-5}$ G_0) observed in Fig. 5a might be due to intrinsic junction-to-junction variation in the details of the molecule/gold electrode contact geometry (J. Am. Chem. Soc. 128, 15874–15881 (2006) and J. Am. Chem. Soc. 128, 2135–2141 (2006) or in some instances, to the formation of multi-molecular bridges. Other sources of conductance variation can be attributed to 1) the uncertainty in the stereochemistry of the silylated olefin (reaction of Si-H with 1-alkynes). The produced silylated olefin could be either trans or cis (J. Am. Chem. Soc. 2011, 133, 4998–5008) which could have different conductivities and 2) the different bias applied (-0.8V for silicon versus -0.2V for gold to maintain the current for both cases in the ca. 3 nA range) which can induce more fluctuation in the silicon case. 1D histograms representing the 2D maps of figure 5a is now added to the SI figure S1.4.1b

Second, in the discussion of the energy band profile in the semiconductor the authors state that they used a Poisson distribution in the semiconductor. This is unclear. Perhaps they mean a linear electric field profile coupled with Poisson's equation, but if they really mean a Poisson distribution, this discussion needs to be expanded.

This was a wrong description that has been now corrected in page 6 line 10: “Solving the Poisson's equation that describes the variation of the electric field across the semiconductor”

Third, the discussion of switching frequency should be described in more detail. Intuitively, I would expect the opposite: the increased mobility would increase the switching probability, not decrease it.

This is an interesting point. We speculate that due to the rigidity of the Si-C bond and the increased stability of the molecular junction, the same molecule could break from the gold electrode while it is still attached to the silicon electrode which gives the possibility of the same molecule capable of forming consecutive junctions. We did observe multiple junctions in relatively a short interval of time in the case of silicon surface (e.g. Fig.S1.4.2 of the SI).

We have added page 7 line 20:

“This enhanced junction formation is attributed to the absence of mobility of the silicon atoms as opposed to that high mobility of gold atoms in gold electrodes. This is supported by occasionally observing more than one blink appearing in a 3 second frame (35 % the blinks in the silicon case appeared accompanied by other blinks versus 5 % in the gold case). Figure S1.4.2 represents an example of a high-frequency blinking capture in a 3 second frame. We speculate that the non-homogenous and sudden increase in the frequency of the junction in the silicon case is attributed to the rigidity of the carbon-silicon bond which enables the same single molecule to form a junction

multiple times in short period of time (breaking from the weak gold electrode and then reforming again) which lead to a sudden increase in the blinking frequency.”

-Finally, changes in dispersion are typically quantified by producing conductance histograms. Although this information is contained in their time blink histograms, it is not as obvious in this presentation.

Conductance histogram is now added as figure S1.4.1b of the SI.

Reviewer #5

I fully agree with the critical concerns by reviewer #1, especially those with the interpretation of the statistical analysis under varying conditions. For example, the different bias cannot justify the results and the corresponding conclusions. In summary, the paper should be rejected.

It appears to us that the reviewer basis his/her unjustified and extremely brief opinion on that provided by Reviewer 1 who recommends a revised version of the manuscript for Nature Comm. We have addressed the remarks by reviewer 5 in our detailed response to reviewer 1. As stated in our response to reviewer 1, we have removed any ambiguity about the presentation of statistics by comparing the coefficient of variation (C_v) of 35 independent curves. We note the bias is not different and it is fixed in all IV experiments, even though the bias would not affect the IV curves at all as we show in Figure S1.1.4. We have stated this clearly in the caption of Figure 4 ““under a surface bias of -0.8 V” and have clearly shown the starting and ending bias (-0.8 V) in Figure S1.1.3. We have also added Figure S1.1.4 that shows that the IV curves do not change when the working bias is changed.

Point-by-point responses to Referee 1:

Referee 1: The authors modified the manuscript properly in general and added statistical analysis to show the reproducibility of the sing-molecular junction on Si substrates. In the present manuscript, however, there seems to be still a few issues about precise description, which I think should be adequately addressed for the publication.

1) All the conditions used should be clearly described. Is the initial set-point current -1 nA throughout the work? If not, please include all the information since some information might be missing due to the vertical offset used in the data plot; some figure options include the value for set-point current, but some do not. If the use of different set-point currents is necessary, is it due to instability of the molecular system or AFM?

Reply: The setpoint current is 1 nA throughout the work. We have stated this in the captions of Figures 5 and 6 in the previous revised manuscript. We now further added in the "final revised manuscript" that the current setpoint is 1 nA in figures 3 and 4. The setpoint current used does not affect the stability of the molecular systems as we have demonstrated in Figure 3 where we check that the molecular junctions are intact after applying the voltage ramp.

2) It is a little surprising to me that the authors stated that the previous Figure 2 was meant to be a schematic and they plotted -0.4 V wrong instead of the actual value -1.6 V. I would like to strongly request the authors to check if all the data and parameters are shown correctly. It is not clear to me why it was not possible to plot real voltages for the new data in Fig. 2 (b) and (c). If some of the data were lost, they should not be used.

Reply: We have stated in the last version that the bottom panels of figure 2b and 2c are real inputs data but are not measured output signals (therefore are not measured data but a guidance to the eye to show how the set bias voltage evolve during the experiment). To clarify we have replaced the y axis of the bottom panels in figure 2b and 2c from "surface bias" to "surface set-bias" and added in the caption of Figure 2 "Surface-set bias plots of panels b and c represent the evolution of the working bias voltage applied between the STM Au tip and the Si substrate".